# Bone Remodelling, Vitamin D Status, and Lifestyle Factors in Spanish Vegans, Lacto-Ovo Vegetarians, and Omnivores

**DOI:** 10.3390/nu16030448

**Published:** 2024-02-02

**Authors:** Elena García-Maldonado, Angélica Gallego-Narbón, Belén Zapatera, Alexandra Alcorta, Miriam Martínez-Suárez, M. Pilar Vaquero

**Affiliations:** 1Institute of Food Science, Technology and Nutrition (ICTAN-CSIC), C. José Antonio Novais 10, 28040 Madrid, Spain; e.garcia@ictan.csic.es (E.G.-M.); angelica.gallego@uam.es (A.G.-N.); bzapatera@ictan.csic.es (B.Z.); alexandra.alcorta95@gmail.com (A.A.); msuarez.miriam@gmail.com (M.M.-S.); 2Biology Department, Universidad Autónoma de Madrid, Darwin, 2, 28049 Madrid, Spain

**Keywords:** vegetarian, vegan, vitamin D status, bone, bone formation, bone resorption, parathormone, physical activity, body composition, nutrient intake

## Abstract

Sustainable healthy diets are promoted, and consequently vegetarian diets are currently increasing. However, scientific information on their effects on bone health is scarce. A cross-sectional study was performed in adults (66% women) classified into three groups: omnivores (*n* = 93), lacto-ovo vegetarians (*n* = 96), and vegans (*n* = 112). Nutrient intake, body composition, physical activity, vitamin D status (25-hydroxycholecalciferol, 25-OHD), parathormone (PTH), and bone formation (bone alkaline phosphatase, BAP) and resorption (N-telopeptides of type I collagen, NTx) markers were determined. Lacto-ovo vegetarians and especially vegans showed lower protein, fat, calcium, phosphorous, vitamin D, retinol, iodine, and zinc intakes, and higher carbohydrate, fibre, carotenes, magnesium, and vitamin K intakes compared to omnivores. Body composition was similar in the three groups that performed vigorous physical activity regularly. Body bone mass and muscle mass were positively correlated with BAP, and time performing physical activity with 25-OHD. The prevalence of vitamin D deficiency or insufficiency (25-OHD < 75 nmol/L) was 93.7% in the studied population, and vitamin D deficiency (25-OHD < 25 nmol/L) was significantly higher in vegans. Vegetarians of both groups had increased PTH and NTx with vegans showing significantly higher PTH and NTx than omnivores. Conclusion: Adult vegetarians, especially vegans, should reduce the risk of bone loss by appropriate diet planning and vitamin D supplementation.

## 1. Introduction

Plant-based diets are increasingly popular in high-income countries. Current recommendations for a healthy and sustainable diet promote increasing the consumption of vegetables, grains, and fruits, and decreasing the intake of processed meat while consuming small amounts of lean meat and fish [1]. In this line, the choice of no consumption of any food of animal origin is also generally accepted provided that the diet is properly planned to prevent nutrient deficiencies [2]. Among the effects of consuming vegetarian diets, a reduction in cholesterol levels and increased insulin sensitivity are well documented [3], both being factors related to low cardiovascular and diabetes risks. Consistently, several reports indicate that mortality from coronary heart disease, diabetes, and kidney disease is lower in vegetarians than in non-vegetarians [4]. However, unhealthy plant-based diets are associated with higher total and disease-specific mortality rates [5].

Previously, research on the nutritional status of Spanish vegetarians found that about two-thirds of them used supplements of vitamin B12, which is only present in food of animal origin [6]. However, the use of other supplements was negligible. Regarding iron status, although iron deficiency anaemia was not frequent, approximately one-third of the participants presented iron deficiency or were iron-depleted, mostly women [7,8]. In contrast, folate status, as measured by levels in erythrocytes, was high [6].

Concerning bone health, there are a number of components of the vegetarian diets that may play a role. On one hand, plant-based diets generally contain lower amounts or less bioavailable nutrients that are essential for bone health, such as protein, vitamin D, calcium, vitamin A, zinc, selenium, and iodine, and on the other hand, they contain more quantities of several protective nutrients, including magnesium, potassium, vitamin K, vitamin C, and phytobioactive compounds [9,10,11,12,13]. In addition, lower animal protein and phosphorous intakes and higher magnesium and potassium intakes are associated with lower acid load and a potential alkalinising effect [14], which has been suggested to be beneficial for bone health, as bone can reduce its buffering activity to maintain acid–base balance and therefore bone minerals are retained. Nevertheless, the scarce studies focused on the relationship between vegetarian diets and bone health suggest that these diets may increase the risk of osteoporosis and fractures. Lower bone mineral density at the femoral neck and lumbar spine has been found in vegetarians compared to omnivores, and the difference was greater for vegans [9]. In the EPIC-Oxford prospective cohort, a higher fracture rate was observed among vegans with low calcium intakes compared to vegetarians who consumed dairy and eggs, fish eaters, and meat eaters [15].

Bone remodelling refers to the process of coordinated bone formation and bone resorption, by the action of the cell’s osteoclasts, osteoblasts, and osteocytes, to renew old or damaged bone and achieve bone mass equilibrium, thus maintaining the strength and biochemical functions of bone. Therefore, healthy adult bones are characterised by a balance between bone formation and resorption. The main regulators of bone remodelling in adulthood are vitamin D and parathormone (PTH) [16,17]. Under situations of vitamin D deficiency, PTH secretion increases to enhance vitamin D metabolism by activating the renal conversion of 25-hydroxycholecalciferol (25-OHD, the biomarker of vitamin D status) to 1,25-hydroxycholecalciferol, which facilitates intestinal absorption of calcium and phosphorous and bone resorption, and at the kidney level, increases phosphorous reabsorption [18].

The biochemical markers of bone remodelling provide information on osteoblast and osteoclast activity. Among the formation markers, bone alkaline phosphatase (BAP) and procollagen I amino-terminal propeptide (PINP) are measured in serum and are very sensitive to changes in bone formation after only several weeks of nutritional intervention. The cross-linked N-telopeptides of type I collagen (NTx) is a specific biochemical marker metabolically derived from osteoclast collagen, whose urine concentration is directly related to the extent of bone resorption. In aged bone, collagen suffers isomerisation, and the β-carboxy-terminal telopeptide of collagen (β-CTX) is the resorption marker of choice, for example in postmenopausal osteoporosis [18,19,20].

In addition, body weight and body composition also play a relevant role, and adequate physical activity is beneficial for optimal muscle and bone interaction, helping to prevent fractures. In this respect, lower body mass index (BMI) was found in vegans compared with meat eaters, but not with fish-eaters or other vegetarians, in the EPIC-Oxford study [21]. In other studies, no differences in BMI or body composition were attributed to vegetarian diets, and two studies reported higher physical activity in vegetarians than omnivores [22,23].

The effects of vegetarian diets on bone health have been studied in different populations, however in Spain, a country where the Mediterranean diet is widely followed and promoted, the scientific information is very scarce. One survey found lower calcium, vitamin D, and iodine intakes in Spanish vegans compared to the general population, but the study did not include any biochemical determinations [24]. In our investigations, we have studied the nutritional status of vegetarians, disaggregating these into two subgroups, lacto-ovo vegetarians and vegans, that have been compared with omnivores. We believe that this approach is useful to obtain highly applicable results, as these diets can be thoroughly characterised and are followed by an increasing number of people.

With this background, the objectives of the present study are (1) to determine if the intakes of nutrients related to bone health vary between Spanish vegetarians and omnivores and if there are differences between lacto-ovo vegetarians and vegans; (2) to know the prevalence of vitamin D deficiency, according to the serum marker 25-OHD, in the three diet groups; (3) to determine PTH and bone remodelling according to diet and gender; and (4) to evaluate if body composition and/or physical activity are interacting factors in the relationship between diet and bone remodelling in Spanish omnivores, lacto-ovo vegetarians, or vegans.

## 2. Materials and Methods

### 2.1. Study Design and Participants

The design is a cross-sectional study. Participants whose usual diet was lacto-ovo vegetarian, vegan, or omnivorous were invited to participate. Recruitment and collection of samples and data were completed in the Madrid region of Spain (latitude 40°24′59.4″ N), during June–July 2017, February 2020, and June–July 2021. Inclusion criteria were healthy adult, omnivore, lacto-ovo vegetarian, or vegan following his/her current diet for at least 6 months. Exclusion criteria were minor (age < 18 years), following his/her current diet for less than 6 months, eating disorders, diagnosis of digestive, haematological, endocrine, renal, or oncological diseases, pregnancy, breastfeeding, menopause, having donated blood in the 3 months prior to the study, and for the lacto-ovo vegetarian and vegan participants occasional consumption of meat or fish was an exclusion criterion. Flexitarians or self-defined vegetarians who consumed some or sometimes meat or fish were excluded while omnivores were considered those consuming both fish and meat in their diets.

A total of 301 subjects (66% women) were recruited and accepted to participate in the study, including 93 omnivores, 96 lacto-ovo vegetarians, and 112 vegans. The distribution of participants according to time of recruitment was *n* = 104 in 2017 (summer), comprising *n* = 49 lacto-ovo vegetarians and 54 vegans; *n* = 99 in 2020 (winter), comprising *n* = 47 omnivores, *n* = 25 lacto-ovo vegetarians, *n* = 27 vegans; and *n* = 98 in 2021 (summer), comprising *n* = 46 omnivores, *n* = 22 lacto-ovo vegetarians, and *n* = 30 vegans.

The study was approved by the Ethics Committees of CSIC (Cod. 104/2019, date 19 December 2019 and modification of Cod. 46/2021, date 5 March 2021) and the Hospital Puerta de Hierro, Majadahonda, Spain (Cod. PI 176/19, date 18 November 2019, and modification of Cod. PI 176/19 date 2 March 2021). Each participant signed an informed consent form before the study began.

### 2.2. Anthropometry and Body Composition

Body weight, height, and waist and hip perimeters were measured by a trained member of the research group by standardised procedures, and body mass index (BMI) was calculated. Bone mass, muscle mass, and percentages of body water and body fat were determined by a body composition monitor (Tanita BC-601, Tanita Ltd., Amsterdam, The Netherlands).

Accuracy of the body weight measurement was certified by international standard procedures by ENAC (Madrid, Spain) according to International Laboratories Accreditation Collaboration. Tolerance was 0.100 kg. Precision was determined by comparing the measures of body weight, muscle mass, and body water of two monitors of the same brand and model. The values were obtained from 6 measures, and frequency of measurement was every year. Results of precision were always <0.1%.

### 2.3. Physical Activity and Dietary Intake Assessments

Physical activity was assessed using the short version of the International Physical Activity Questionnaire (IPAQ). Data are presented as vigorous, moderate, walking, and sitting activities in min/week and total metabolic equivalent of task (MET minutes/week). According to guidelines by WHO, infographic sheets and examples of vigorous and moderate activity were given [25].

Dietary intake was assessed by 72 h dietary records, that were collected in February 2020 and June–July 2021, winter-to-summer ratio of 1:1 (*n* = 207, 62% women). Participants were instructed to fill out an online questionnaire providing details of all foods eaten, the type or brand of the products and the portion or weight of the food consumed, as well as their consumption of supplements. They were required to write down this information just after finishing their meal and they could provide photos or labels of the foods they ingested and ask questions by phone or email. Data were analysed using the DIAL software, version 3.15 (Alce Ingeniería, Madrid, Spain). This database was extended by adding the nutritional composition of vegan products, using information from the manufacturer.

Protocols and procedures fulfil the standard ISO 9001:2015 [26].

### 2.4. Sampling and Analytical Determinations

Participants attended the Human Nutrition Unit of the institute between 7:45 h and 9:30 h after a 10–12 h fasting period. They were asked a few questions in order to check their health status and smoking habits. Blood samples were collected by venepuncture and serum was obtained after centrifugation at 1000 g for 15 min. The fasting second-morning urine was also collected to analyse creatinine and the bone resorption marker aminoterminal telopeptide of collagen I (NTx).

Serum 25-hydroxyvitamin D (25-OHD) and the bone formation marker (bone alkaline phosphatase, BAP), were determined by ELISA kits 25-hydroxyvitamin D EIA and Ostase BAP EIA, commercialised by Immunodiagnostic Systems Holdings (IDS, Boldon, UK); parathyroid hormone (PTH) was analysed by ELISA using the kit PTH Parathyroid Intact EIA from DRG (DRG Instruments GmbH, Marburg, Germany); and the bone resorption marker NTx was determined using the ELISA kit NTx Urine Osteomark (Alere Scarbourgh Inc., Scarborough, ME, USA). The intra- and inter-assay coefficients of variation were as follows: PTH, 2.7 and 4.1%; 25-OHD, 2.3% and 8.2%; BAP, 2.8% and 6.4%; and NTX, 4.1% and 3.4%. All bone turnover markers were analysed using one single batch for each marker. Urine creatinine was measured using an autoanalyser method (Olympus AU5800, Beckman, Nyon, Switzerland), with intra- and inter-assay coefficients <1 and <2%, respectively. NTx is expressed as nanomoles of bone collagen equivalents per millimole of creatinine (nM BCE/mM Cr). All determinations comply with the standard ISO 9001:2015 requirements.

### 2.5. Vitamin D Status Levels

Vitamin D status of the volunteers was studied considering the following cut-off values of serum 25-OHD [27,28]: >75 nmol/L (>30 ng/mL), sufficiency; <75 and >50 nmol/L (<30 and >20 ng/mL), insufficiency; <50 and >25 nmol/L (<20 ng/mL and >10 ng/mL), deficiency; and <25 nmol/L (<10 ng/mL), severe deficiency.

### 2.6. Statistical Analyses

Variable normality was evaluated by the Kolmogorov–Smirnov test and visually and if possible non-normally distributed variables were log-transformed for normalisation. Variables that could not be normalised were analysed by non-parametric tests. Data are presented as mean ± SD for normalised variables or median and 95% CI in case of non-normal variables.

General linear models were used to test the effects of the factors of dietary group (omnivores, lacto-ovo vegetarians, and vegans), gender (woman, man), season (winter, summer), and the group–gender and group–season interactions. Further models included age as covariate. The Bonferroni correction for multiple comparisons was used. Physical activity and micronutrient intakes were analysed by the non-parametric Kruskal–Wallis (to compare 3 groups) and Mann–Whitney tests (to compare 2 groups). The relationship between selected variables was studied by Pearson or Spearman correlations, depending on the parametric or non-parametric distributions, respectively. Prevalence of the different levels of vitamin D status (severe deficiency, deficiency, insufficiency, and sufficiency) was tested by Pearson chi-square tests, and pairwise comparisons of the proportions were made using the Bonferroni correction.

The data were analysed using SPSS for Windows version 26.0 (IBM SPSS Statistics for Windows, Armonk, NY, USA). The level of significance was set at *p* < 0.05.

## 3. Results

The characteristics of the participants are presented in Table 1. There were no significant differences between the three groups in body weight, BMI, waist and hip perimeters or in the selected body composition parameters, bone mass, muscle mass, and the percentages of body fat and water. Age presented significant differences, with lacto-ovo vegetarians and vegans being significantly older than omnivores. All anthropometric and body composition parameters show significant differences between men and women.

Characteristics of participants whose data of nutrient intake and bone parameters are available did not vary from those of the total sample, ages were (mean ± SD): 26 ± 6, 25 ± 5, and 28 ± 6 year, for omnivores, lacto-ovo vegetarians, and vegans, respectively.

Figure 1 shows the usual physical activity performed by the volunteers. Total activity did not present significant differences between dietary groups (median values: 2565, 3127, and 2663 MET min/week for omnivores, lacto-ovo vegetarians, and vegans, respectively). Lacto-ovo vegetarians performed significantly higher moderate activity than omnivores, but no other group differences were detected. Sitting time was similar in the three groups, with a median value of 420 min per day (7 h).

The subsample of participants of which nutrient intake and body remodelling markers are available also shows significantly higher moderate activity in lacto-ovo vegetarians compared to omnivores.

Results show that higher vigorous activity was performed by men (*p* = 0.046) while women did more walking (*p* = 0.013).

Among other lifestyle factors, smoking was considered. It was estimated that 11%, 11%, and 8% of the omnivores, lacto-ovo vegetarians, and vegans, respectively, were current smokers.

The consumption of supplements apart from vitamin B12 was low, as less than 3% of the total participants took calcium or vitamin D supplements. 

Regarding dietary intake (Table 2), total energy intake did not differ between groups, but there were profound differences in the macronutrient contribution to the energy and in the fibre intake. The energy provided by proteins was significantly lower in lacto-ovo vegetarians and vegans than omnivores (*p* < 0.001), and in contrast, carbohydrate contribution was significantly higher, particularly in vegans, with significant differences between the three groups (*p* < 0.001). Fibre intake was the highest in vegans, followed by lacto-ovo vegetarians and omnivores (*p* < 0.001). Women presented lower total energy and fibre intake compared with men (*p* < 0.001). Fat intake, approximately 40% of the energy in omnivores and lacto-ovo vegetarians, was significantly lower in vegans (*p* < 0.001). Concerning the type of fat ingested, the percentage of saturated fat consumed was lower in lacto-ovo vegetarians and vegans than in omnivores, with significant differences between the three groups. Moreover, vegans ingested significantly less monounsaturated and polyunsaturated fat than omnivores. 

Calcium intake was lower in vegans than in lacto-ovo vegetarians and omnivores (*p* < 0.001), while phosphorous intake was only significantly lower in vegans when compared to omnivores (*p* = 0.005). However, the values of magnesium intake were higher in lacto-ovo vegetarians and vegans compared to omnivores (*p* < 0.001). No differences in potassium or sodium intake were detected. Iodine intake was lower in the two vegetarian groups than in omnivores, and that of zinc in vegans compared to omnivores. Mineral intake was significantly lower in women than men in all cases. 

Regarding vitamin intakes, vitamin D intake was significantly lower in vegetarians of both groups compared to omnivores (*p* < 0.001), but vitamin K intake was higher, especially in vegan men when compared to omnivorous men (*p* = 0.003). Vitamin C intake was higher in vegetarians than omnivores (*p* = 0.037), although pair differences did not reach the significance level. Vitamin A consumption was also higher in the two vegetarian groups (*p* = 0.003), and the contribution of carotenes explained the differences, as retinol intake was lower in vegetarians than omnivores and very low in vegans (*p* < 0.001). Vitamin E consumption was slightly higher in the two vegetarian groups compared with omnivores (*p* = 0.012). Women ingested lower amounts of vitamins E and K than men.

Vitamin D status is shown in Figure 2. The prevalence of severe vitamin D deficiency, 25-OHD < 25 nmol/L (10 ng/mL), was higher in vegans compared to lacto-ovo vegetarians and omnivores. The prevalence of vitamin D deficiency, insufficiency, or sufficiency did not vary among groups.

Table 3 shows PTH, 25-OHD, and the two remodelling markers, BAP and NTx, according to group and gender. There were group differences in PTH and NTx. PTH was significantly higher in vegans than omnivores (*p* = 0.003), while NTx was higher in lacto-ovo vegetarians and vegans (*p* = 0.002) without significant pair differences. Values of 25-OHD tended to decline, being the highest in omnivores, and the lowest in vegans but did not present significant group or gender differences. Women presented higher BAP than men (*p* < 0.001).

The concentrations of creatinine in urine were (mean ± SD): 152 ± 87, 147 ± 105, and 127 ± 70 mg/dL, without significant differences between groups (*p* = 0.223).

The study of the associations between variables confirms that BAP and NTx are positively correlated (r = 0.248, *p* = 0.001, *n* = 183). In addition, PTH is also related to NTx (r = 0.206, *p* = 0.005, *n* = 183) while there is a negative association between PTH and 25-OHD (r = −0.360, *p* < 0.001, *n* = 183). Vitamin D intake was not associated with any parameter of bone metabolism; however, higher 25-OHD levels were obtained in summer compared to winter (general linear model, *p* < 0.001).

Physical activity and 25-OHD were related, as sitting was negatively correlated with 25-OHD (rho = −0.204, *p* = 0.006, *n* = 178). No other significantly relevant associations were found.

## 4. Discussion

In this study, bone remodelling is analysed in Spanish vegans, lacto-ovo vegetarians, and omnivores and the possible influence of dietary intake and physical activity is evaluated. It was found that both PTH, which is a strong regulator of bone metabolism, and the bone resorption marker, NTx, are increased in the two groups of vegetarians, although lacto-ovo vegetarians appear to be more protected from bone loss.

These participants were very well characterised, all lived in the Madrid region of Spain, and it was checked that they were correctly classified in one of the diet groups. At first, the obtained results can be explained by the lower intakes of protein, calcium, phosphate, zinc, and vitamins A and D in the vegan group. However, lacto-ovo vegetarians also ingested lower amounts of several nutrients than omnivores, for example, protein and vitamin D, and the influence on bone parameters appeared to be lower.

This can be explained considering that lacto-ovo vegetarians ingested protein from animal products, dairy, and eggs, and more calcium than vegans. In addition, calcium from the lacto-ovo vegetarian diet probably had high bioavailability, because the amount of fibre provided with this diet was not as high, and it is known that there are inhibitors of calcium absorption, such as oxalates and phytates associated with fibre. Moreover, the higher vitamin D intake in lacto-ovo vegetarians than in vegans, although inadequate in both groups, may have also contributed to calcium absorption and the maintenance of bone remodelling.

Our results on the dietary intake of vegetarians agree with the available data from Spanish studies concerning macronutrients, calcium, and vitamin D [24]. We previously presented the differences in macronutrient intakes, lower intakes of proteins and saturated fat and at the same time higher carbohydrate and fibre intakes in lacto-ovo vegetarians and especially in vegans compared to omnivores [8,22]. However, one study shows in vegetarians (including vegans), calcium intake is much higher than in the present study, with mean values above the level of 1000 mg per day [29]. The differences may be due to the method used for dietary assessment, as we used 72 h dietary registries while in that study a food frequency questionnaire was applied. In the EPIC-Oxford cohort study [30], which included 18,840 lacto-ovo vegetarians and 2596 vegans, it was found that vegans ingested less calcium than other vegetarians, meat eaters, and fish eaters, in line with our findings. Moreover, the mean values of calcium intake reported for this subgroup, 610 (SD 241) and 582 (SD 242) mg/day, for men and women, respectively, were similar to those of the present study.

Very low vitamin D intakes in vegetarians, and extremely low in vegans have been consistently presented in the literature [24,29,30,31,32]. It should be noted that the main source of vitamin D is exposure to sunlight and that during winter this mechanism is generally not sufficient for vitamin D demands; therefore, a body storage of vitamin D or supplemental consumption is needed. In this regard, the analysis of serum 25-OHD confirms that there was a minimum level in winter for all studied groups. Overall, the status of this vitamin was insufficient or deficient in most of the subjects, as more than half of our sample presented 25-OHD levels below the cut-off of 50 nmol/L (<20 ng/mL), and most importantly, the prevalence of severe vitamin D deficiency, 25-OHD < 25 nmol/L (<10 ng/dL) was significantly higher in vegans compared to the other groups. It is remarkable that only 6.3% of our studied population presented sufficient vitamin D status and that there were small differences among the diet groups.

The paradox of low vitamin D status in Mediterranean countries such as Spain has been investigated before. Although it is a sunny country, factors like avoiding the heat and urbanisation that involves indoor activities may play an important role [32]. In this regard, our results show that total vitamin D intake was much lower than the 15 µg/day required to prevent vitamin D deficiency (25-OHD < 30 nmol/L) when sunshine exposure is limited [33]. In previous studies by our research group, performed in the same geographic region as the present study, we consistently observed low vitamin D intakes, with average values around 3 µg/day and serum 25-OHD levels between 50 and 75 nmol/L [19,34]. 

To our knowledge, there are no published data on 25-OHD in Spanish vegetarians previous to this study. However, our results are in agreement with a study on Danish vegans [35], which showed a higher prevalence of vitamin D insufficiency and deficiency in vegans in comparison to omnivores. However, the values of 25-OHD in our study were lower, which should be interpreted considering the differences in supplementation, as 59% of the Danish vegans took vitamin D supplements while this practice was scarce in Spanish vegans. Another report on German adolescents [36] also shows low vitamin D status in vegetarians and vegans. In fact, our 25-OH levels are lower than the values reported by Neufingerl and Elander [37] in their systematic review. They obtained average values lying in the insufficient vitamin D status category while our data are in the deficiency range. Therefore, we must point out that participants of the present study were not aware of their risk of vitamin D deficiency, as the use of supplements containing this vitamin was very uncommon. In this regard, Spanish vegans may regard vitamin D supplementation as unnecessary given the sunny conditions of the country. Moreover, form D2 (ergocalciferol) of vegetal origin appears to be less bioavailable than form D3 of animal origin [38], which should be known by vegetarians to adjust appropriate supplementation doses. 

The high intake of vitamin K might partly compensate for the effects of the low intake of calcium and vitamin D in vegetarians since the estimated dairy intake was above the recommended amounts [39]. Concerning vitamin A, retinol was almost absent in the vegan diet, but the contribution of carotenes, with known vitamin A activity, could play a protecting role that cannot be underestimated.

Regarding the results of bone parameters, the PTH obtained for vegans was at the cut-off level of hyperparathyroidism, which has been set at 65 pg/mL [40], and although there are no standardised cut-off values for BAP and NTx, the obtained BAP values are similar to those reported by Hansen et al. [35] in young vegan and vegetarians, and NTx levels are above [19] or similar to those found in previous studies on young women [34]. Therefore, in the present study, PTH secretion is needed, and both bone formation and resorption are promoted, resulting in higher resorption in vegetarians than in omnivores, especially in the vegan group, which presented the lowest intakes of proteins, calcium, and vitamin D, nutrients related to bone health. This also suggests that vegetarians, especially vegans, are at risk of secondary hyperparathyroidism, a situation that enhances bone resorption. Concerning the methods of analyses used, NTx was measured in urine and the values were calculated as NTx/creatinine (nM/mM). Omnivores tended to excrete higher amounts of creatinine than vegetarians, as animal food is a source of creatinine, and although creatinine concentrations did not show significant differences, this should be considered as a possible confounder.

Low zinc and iodine intakes have also been reported in vegetarians [11,13]. These minerals participate in cellular growth and their deficiencies could affect bone health [41,42]. The strategies to improve their status include the use of iodine salt and proper cooking techniques to favour phytate hydrolysis, which could decrease zinc binding and enhance its absorption [4].

The possible implications of iron deficiency on bone health have been suggested in previous research [19,34]. Iron plays a role in collagen synthesis, and collagen constitutes 90% of total bone proteins, and participates in the activation of vitamin D, as the 25-, 1-alfa-, and 24-hydroxylases are all haem-enzymes. In this regard, we previously reported that plant-based diets may induce a lower iron status than omnivorous diets [7,8], supporting the idea of a relationship between iron status and bone health in vegetarians.

Another relevant aspect that we have evaluated is the possible interaction factor of the physical activity performed by the volunteers. Generally, all three groups performed physical activity regularly and were classified as active. However, lacto-ovo vegetarians performed more moderate activity than omnivores, which could have contributed to protecting their bones. In this regard, we observed that 25-OHD was positively associated with total physical activity and negatively with the time spent sitting. 

Overall, the findings of the present study point to a higher risk of bone loss in vegans compared to omnivores. Certainly, calcium and vitamin D intakes should be increased. In addition, the role of protein concerning the bone health of vegetarians should not be undervalued. Protein digestibility has been extensively studied and the quality of plant protein, such as soya protein, is generally high. However, the amount of protein is crucial for the maintenance of bone and muscle mass in adulthood [43] and to prevent osteoporosis, as well as sarcopenia in the elderly. In this regard, the European Society on Parenteral and Enteral Nutrition (ESPEN) recently recommended increasing the intake of protein to 1.0–1.5 g per kg body weight per day to retard age-related muscle loss [44]. Average ratios in the present study are 1.3, 1.3, and 1.1 in the intake of omnivores, lacto-ovo vegetarians, and vegans, respectively. Therefore, vegans should be aware of the importance of consuming sufficient amounts of high-quality protein. 

There are limitations of this study that need to be mentioned. First of all, this is a cross-sectional study, and no relationship between cause and effect can be derived from it. Secondly, nutrient intake assessment and bone remodelling determinations were only available from two-thirds of the studied population. However, the characteristics of this subsample were similar to the whole sample in terms of age, body composition, and physical activity behaviour, and the possible confounder effect of age was considered. Thirdly, sun exposure was not known, which could have provided information on the cutaneous synthesis of vitamin D relative to the dietary source.

The fact that the study was completed in Spanish vegetarians is a strength, as there are very few studies that have explored the nutritional status of vegetarians in this country and our findings confirm the Mediterranean paradox, as the high prevalence of vitamin D deficiency is outstanding.

## 5. Conclusions

Plant-based diets supply high quantities of bone-protective phytochemicals, but the limited amount and bioavailability of the very well-known dietary factors, protein, calcium, and vitamin D, appear to be crucial in our study. The study presented here shows a generally low vitamin D status in Spanish lacto-ovo vegetarians and vegans but also in omnivores, with vegans presenting a higher prevalence of severe vitamin D deficiency. This situation determines PTH elevation and bone resorption, which in the long term could induce bone loss. Therefore, it is important that consumers of vegetarian diets are aware of this risk and, in addition to performing physical activity, they increase the intake of protein, calcium, phosphorus, and particularly vitamin D.

### Directions for Further Investigations

Further research should be conducted on vegetarians in comparison with omnivores to evaluate changes in bone health during one’s lifespan. In this regard, nutrition during childhood and adolescence is critical to achieving a high peak bone mass in adulthood, sufficient to prevent osteoporosis and associated bone fractures in old age. Given our results, this may be especially important in vegetarians. Prospective cohort studies including the study of lifestyle factors, dietary intake, use of supplements, and disease incidence are very important in this population because undernutrition in the elderly is often a cause of frailty. In addition, the need for protein and micronutrient supplementation should be explored in older adults who are long-term consumers of plant-based diets.

## Figures and Tables

**Figure 1 nutrients-16-00448-f001:**
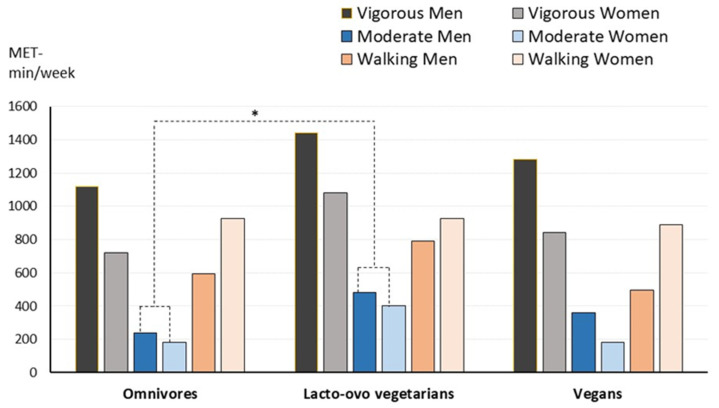
Level of physical activity (vigorous, moderate, or walking) performed by the participants according to dietary group and gender. Median values are shown. Omnivores, *n* = 92, lacto-ovo vegetarians, *n* = 96, and vegans, *n* = 112. * Significantly different compared to omnivores. Men performed significantly higher vigorous activity (*p* = 0.046) and less walking than women (*p* = 0.013). Kruskal–Wallis and Mann–Whitney tests.

**Figure 2 nutrients-16-00448-f002:**
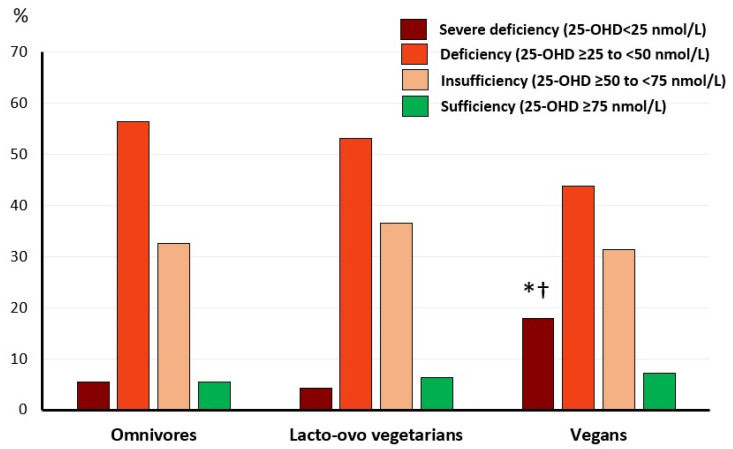
Prevalence of vitamin D deficiency, insufficiency, and sufficiency according to diet. Values are % within group. Omnivores, *n* = 92, lacto-ovo vegetarians, *n* = 96, and vegans, *n* = 112. Proportions differ significantly compared to: * omnivores; ^†^ lacto-ovo vegetarians (Pearson chi-square test, *p* = 0.021).

**Table 1 nutrients-16-00448-t001:** Age, anthropometrics, and body composition of the participants.

	Omnivores (*n* = 93)		Lacto-Ovo Vegetarians (*n* = 96)		Vegans (*n* = 112)		*p* Group	*p* Gender
	Man (*n* = 41)	Woman (*n* = 52)	Man (*n* = 27)	Woman (*n* = 69)	Man (*n* = 32)	Woman (*n* = 80)		
Age	25.8 ± 6.7	25.2 ± 5.1	27.8 ± 7.9 *	28.8 ± 7.3 *	30.6 ± 6.4 *	28.1 ± 6.4 *	<0.001	0.482
Body weight (kg)	72.0 ± 12.2	58.0 ± 7.8	72.9 ± 12.0	58.8 ± 8.2	72.5 ± 9.7	58.5 ± 8.0	0.814	<0.001
BMI (kg/m^2^)	23.2 ± 3.1	21.7 ± 2.3	23.3 ± 3.7	22.3 ± 3.4	23.3 ± 3.1	21.9 ± 2.9	0.765	<0.001
Waist perimeter (cm)	82.6 ± 8.5	73.7 ± 6.4	84.6 ± 11.2	76.2 ± 8.5	84.2 ± 8.5	75.7 ± 8.1	0.193	<0.001
Hip perimeter (cm)	94.9 ± 7.4	94.7 ± 6.4	95.6 ± 6.5	96.1 ± 7.1	94.9 ± 5.6	95.2 ± 6.2	0.809	0.627
Bone mass (kg) ^#^	3.0 (2.9, 3.3)	2.2 (2.2, 2.3)	3.1 (2.8, 3.3)	2.2 (2.2, 2.3)	3.0 (2.9, 3.2)	2.2 (2.2, 2.3)	0.970	<0.001
Muscle mass (kg)	57.6 ± 7.9	41.7 ± 3.5	58.3 ± 8.0	40.9 ± 3.3	58.3 ± 6.2	41.1 ± 3.6	0.994	<0.001
Water (%)	59.6 ± 10.2	56.4 ± 4.4	61.1 ± 5.6	54.6 ± 4.3	60.9 ± 4.7	54.9 ± 4.4	0.952	<0.001
Fat (%)	15.2 ± 5.3	23.7 ± 6.0	15.0 ± 6.8	25.9 ± 6.1	14.9 ± 6.1	25.4 ± 6.1	0.536	<0.001

Values are mean ± SD or median (95% IC). ^#^ Variable log-normalised before statistical analyses. Effects of group and gender are shown. There were no significant group × gender interactions. * Significantly different compared to omnivores (multivariate general linear model and post hoc Bonferroni test).

**Table 2 nutrients-16-00448-t002:** Energy, fibre, macronutrient, mineral, and vitamin intakes (per day).

	Omnivores (*n* = 88)		Lacto-Ovo Vegetarians (*n* = 53)		Vegans (*n* = 59)		*p* Group	*p* Gender
	Man (*n* = 39)	Woman (*n* = 49)	Man (*n* = 16)	Woman (*n* = 37)	Man (*n* = 22)	Woman (*n* = 37)		
Energy (Kcal)	2407 ± 101	1979 ± 661	2716 ± 756	2004 ± 505	2632 ± 619	1955 ± 548	0.306	<0.001
Proteins (%E)	17.3 ± 4.0	16.5 ± 3.7	14.1 ± 4.2 *	14.7 ± 4.0 *	14.0 ± 4.2 *	14.9 ± 4.6 *	<0.001	0.721
CHO (%E)	38.1 ± 7.8	37.6 ± 8.4	41.6 ± 6.0 *	43.5 ± 6.8 *	45.7 ± 10.4 *^,†^	47.3 ± 8.4 *^,†^	<0.001	0.418
Fibre (g)	28.3 ± 14.3	24.6 ± 12.0	50.2 ± 23.6 *	32.7 ± 12.4 *	57.7 ± 33.2 *^,†^	39.0 ± 14.0 *^,†^	<0.001	<0.001
Fat (%E)	40.0 ± 8.3	42.1 ± 9.7	39.8 ± 6.8	37.1 ± 7.9	35.1 ± 9.1 *^,†^	32.8 ± 9.2 *^,†^	<0.001	0.486
SFA (%E)	12.4 ± 2.9	12.7 ± 4.0	9.4 ± 3.5 *	10.6 ± 3.9 *	6.0 ± 1.9 *^,†^	7.2 ± 3.3 *^,†^	<0.001	0.087
MUFA (%E)	16.1 ± 4.3	17.9 ± 6.2	16.4 ± 5.4	14.3 ± 6.0	15.2 ± 6.2 *	12.8 ± 5.5 *	0.011	0.297
PUFA (%E)	5.8 ± 2.9	6.0 ± 2.3	6.8 ± 2.0	5.9 ± 2.7	8.2 ± 6.5 *	6.8 ± 3.0 *	0.020	0.201
Ca (mg)	878 ± 361	759 ± 327	851 ± 352	703 ± 427	644 ± 223 *^,†^	559 ± 259 *^,†^	<0.001	0.022
P (g)	1.63 ± 0.61	1.30 ± 0.47	1.68 ± 0.85	1.20 ± 0.48	1.40 ± 0.50 *	0.95 ± 0.42 *	0.005	<0.001
Mg (mg)	353 ± 144	280 ± 106	519 ± 269 *	369 ± 152 *	512 ± 176 *	364 ± 184 *	<0.001	<0.001
Na (g)	2.46 ± 0.12	1.97 ± 0.78	2.22 ± 0.98	2.01 ± 1.03	2.14 ± 1.29	1.53 ± 0.89	0.090	0.005
K (g)	3.32 ± 0.13	2.78 ± 1.00	4.09 ± 1.56	3.07 ± 1.28	3.82 ± 1.55	2.92 ± 1.20	0.061	<0.001
I (µg) ^#^	100 (81, 112)	74 (63, 88)	43 (32, 102) *	47 (34, 57) *	56 (43, 68) *	38 (32, 39) *	<0.001	<0.001
Zn (mg) ^#^	11.0 (10.2, 13.7)	8.3 (7.8, 9.0)	9.6 (7.6, 14.7)	7.8 (6.8, 8.5)	9.3 (7.6, 11.6) *	6.0 (4.7, 8.2) *	0.006	<0.001
Vit A (RE µg)	628 (584, 716)	705 (563, 873)	1101 (792, 2191) *	923 (699, 1166) *	872 (474, 1117) ^†^	643 (438,873) ^†^	0.016	0.428
Retinol (µg)	337 (279, 379)	234 (186, 292)	267 (75, 359) *	144 (73, 224) *	0.85 (0.0, 10.8) *^,†^	4.5 (0.4, 13.9) *^,†^	<0.001	0.540
Carotenes (mg)	1.4 (1.0, 1.8)	2.0 (1.3, 2.6)	3.1 (2.1, 9.3) *	3.5 (1.5, 5.3) *	3.7 (2.5, 6.0) *	2.7 (2.1, 4.5) *	<0.001	0.597
Vit C (mg)	107 (79, 166)	99 (77, 141)	149 (115, 239)	134 (86, 184)	129 (101, 182)	122 (107, 163)	0.032	0.100
Vit D (µg)	2.2 (1.9, 3.0)	2.1 (1.1, 2.7)	1.4 (1.0, 2.5) *	1.1 (0.5, 1.7) *	0.6 (0.3, 1.4) *	0.9 (0.5, 1.4) *	<0.001	0.480
Vit E (mg)	8.3 (8.0, 10.7)	9.7 (8.1, 11.7)	14.8 (14.5, 18.4)	9.5 (8.0, 11.3)	12.2 (10.4, 16.3)	11.0 (7.5, 14.2)	0.037	0.001
Vit K (µg)	126 (106, 179)	120 (100, 160)	274 (147, 289)	135 (90, 178)	288 (228, 365) *	129 (98, 261)	0.005	0.002

Values are mean ± SD or median (95% CI). Abbreviations: (%E), percentage of the energy; CHO, carbohydrates; SFA, saturated fat; PUFA, polyunsaturated fat; MUFA, monounsaturated fat. ^#^ Variables log-normalised before statistical analyses. The effects of group and gender are shown. The group × gender effects were not significant, except for fibre (*p* = 0.020). * Significantly different compared to omnivores; ^†^ significantly different compared to lacto-ovo vegetarians. Multivariate general linear model and post hoc Bonferroni test, except for vitamin data that were tested by Kruskal–Wallis or Mann–Whitney tests.

**Table 3 nutrients-16-00448-t003:** Serum parathormone, vitamin D status, as serum 25-OHD, bone remodelling, serum BAP, and urine NTx of the participants.

	Omnivores (*n* = 86)		Lacto-Ovo Vegetarians (*n* = 44)		Vegans (*n* = 56)		*p* Group	*p* Gender
	Men (*n* = 38)	Women (*n* = 48)	Men (*n* = 15)	Women (*n* = 29)	Men (*n* = 19)	Women (*n* = 37)		
PTH (pg/mL)	49.2 ± 22.2	49.8 ± 21.0	57.9 ± 15.6	56.6 ± 20.1	59.4 ± 24.1 *	65.8 ± 21.3 *	0.003	0.687
25-OHD (nmol/L)	46.6 ± 16.2	46.4 ± 16.2	42.4 ± 12.8	45.2 ± 15.8	40.7 ± 19.0	38.6 ± 20.3	0.069	0.908
BAP (µg/L) ^#^	20 (18, 22)	15 (14, 16)	19 (12, 24)	14 (11, 19)	18 (14, 22)	15 (14, 18)	0.104	<0.001
NTx (nM/mM creatinine) ^#^	78 (66, 103)	63 (56, 79)	93 (75, 117)	77 (66, 107)	90 (77, 115) *	91 (66, 108) *	0.002	0.136

Values are mean ± SD or median (95% IC). ^#^ Variables log-normalised before statistical analyses. The effects of group and gender are shown. The group × gender effects were not significant. * Significantly different compared to omnivores (multivariate general linear model adjusted for age and post hoc Bonferroni test).

## Data Availability

The data presented in this study are available on request from the corresponding author. The data are not publicly available due to industrial contract agreement.

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
