# Peer review of "Bone Remodelling, Vitamin D Status, and Lifestyle Factors in Spanish Vegans, Lacto-Ovo Vegetarians, and Omnivores"

_nutrients, 2024, doi:10.3390/nu16030448_

Round 1

Reviewer 1 Report

Comments and Suggestions for Authors

The study was well designed, and the manuscript is well-spoken, well-organized, and has adequate results. However, there are two minor flaws that should be added or explained as follows,

1. There was a significant difference in the age of the participants between the different groups, and it is well known that there is a strong relationship between age and serum bone turnover makers, vitamin D and PTH levels. Please add to the discussion how to avoid the influence of the age factor on the conclusions of this study.

2. Regarding bone turnover markers, the International Osteoporosis Foundation and the guidelines related to osteoporosis in different countries recommend serum C-terminal telopeptide type-1 collagen (CTX) and serum carboxyterminal propeptide of type-I collagen (PINP) as bone resorption marker and bone formation marker, respectively, whereas the authors here used urine NTx and serum BAP as bone turnover markers, Why???

Author Response

The study was well designed, and the manuscript is well-spoken, well-organized, and has adequate results. However, there are two minor flaws that should be added or explained as follows,

Thank you for your opinion on our work. We really appreciate it. Bellow you can find our answer to your points.

  1. There was a significant difference in the age of the participants between the different groups, and it is well known that there is a strong relationship between age and serum bone turnover makers, vitamin D and PTH levels. Please add to the discussion how to avoid the influence of the age factor on the conclusions of this study.

Thank you for this observation. We observed that lacto-ovo vegetarians and vegans were older that omnivores. PTH and the vitamin D status marker did not vary with age, but the bone biomarkers presented a significant relationship with age. Both were negatively correlated with age, and we observed that adding age as a covariate in the models did not change the significant results. However, this is now presented in the statistical section, results and discussion.

Action taken: We have added a sentence in the Statistical section: “Further models included age as a covariate” (line 190). Table 6 (in the revised manuscript) has been modified and the statistical results include the age adjustment. Finally, we have added in the conclusion that the possible confounder of age was considered.

  1. Regarding bone turnover markers, the International Osteoporosis Foundation and the guidelines related to osteoporosis in different countries recommend serum C-terminal telopeptide type-1 collagen (CTX) and serum carboxyterminal propeptide of type-I collagen (PINP) as bone resorption marker and bone formation marker, respectively, whereas the authors here used urine NTx and serum BAP as bone turnover markers, Why???.

Thank you for your comment. We agree that it is preferable to use the same markers that other authors. Unfortunately, we did not have access to automatic determinations as recommended by the International Osteoporosis Foundation (IOF) that recommended the analysis of serum PINP and CTX as reference standards for use in observational studies and clinical trials in osteoporosis. But this is not unanimous in bone-related studies. In the present study, we have analysed the bone specific-alkaline phosphatase (and not total alkaline phosphatase) which is highly correlated with P1NP, and urine NTx (collecting the fasting second morning urine), and the same methodology was used to analyse all samples. Action taken: We have included one additional reference in the Introduction (Papakitsou E. Hormones 20, 545–555, 2021. https://doi.org/10.1007/s42000-021-00276-4) that shows the relation of these markers. Moreover, we have included the intra- and inter-assay coefficients of variation of the bone turnover markers in the Methods section (lines 170-172).

Reviewer 2 Report

Comments and Suggestions for Authors

This cross-sectional study comprising omnivores (n=93), lacto-ovo vegetarians (n=96) and vegans (n=112) compared the nutrient intake, body composition, physical activity, vitamin D status, parathormone, and bone formation (bone alkaline phosphatase, BAP) and resorption (N-telopeptides of type I collagen, NTx) among these dietary groups. Although this study might contribute important evidence of nutrient intake, biomarker status and lifestyle factors according to different types of diet, I have concerns and comments.

11. Introduction: Concerning bone health more nutrients need to be mentioned, e.g. selenium, iodine, vitamin A and references in particular with respect to previous studies on vegetarian /vegan diets are missing.

Line 46 in the word „health“ is missing the l

2. Recruitment: Study participants were recruited at 3 time points in 3 different years and different characteristics (e.g. dietary records, biomarkers) were investigated. Please, explain in more detail how many people were recruited at which time point, whether and which differences exist between participants and why different characteristics were investigated.

3.       Inclusion criteria: Why did you decide for 6 month of following the diet as inclusion criteria and not 12 month? What does mean occasional consumption of meat or fish for vegetarians / vegans? Are there any further infromation about the definition of omnivorous diet? Did you have any definition of how often meat or fish has to be consumed?

4.       Methods: Authors mention that the body composition was determined by the valdiated body composition monitor Tanita BC 601. Please, provide more information about the validation oft the Tanita BC 601.

5.       Methods of biomarker measurement: According to biomarker measurements with different ELISA for bone markers intra-and interassay correlation coefficients should be presented.

6.       Method use of spot urine marker: There ist a risk of misinterpretation of biomarker measurements in spot urine adjusted for creatinine in particular in studies comparing plant based diets with omnivores due to the consumption of meat and fish as exogenous sources of creatinine. Therefore, the results and interpretation of differences in NTx between dietary groups might be not correct.

7.       Statististical method: The numbers of included participants change in different tables. Therefore, the results cannot be compared and interpreted together. It is not clear, which people are used for showing the results in different tables. Furthermore, it is not clear how many women and men contribute to different dietary groups. These numbers need to be presented. Since age differs and sex might also differ in different dietary groups, group comparisons should be adjusted at least for age and sex. Also other lifestyle factors like physical activity and smoking (is competely missing so far as lifestyle characteristic and should be presented) might contribute to differences and should be considered.

       8.    Statistical method: How was the Bonferroni correction applied? It was only mentioned in the section statistical methods, but not in tables or results mentioned again.

       9.   Results: As mentioned before numbers of participants vary in different tables and authors should try to show all results for the same population or mention how the results would look like for       those participants with presented biomarker levels. Let me explain in more detail. When you interpret the meaning of physical activity or body composition according to bone markers than also the results for physcal activtiy or body composition should be presented only for participants with measured bone markers.

    10.  Results: It is not clear how many participants were included for the presentation of correlations. Results of correlations are in general overinterpreted. The correlation coefficiants are low (about 0.2-0.4) or irrelevant (<0.2) and should be therefore not be interpeted as strong. Thereby, the p value is not helpful for interpretation.

    11. Results: There are 7 tables and no figures so far presented in the paper. In particular results presented in table 2 and 6 might be possible to be shown in figures?

   12. Discussion: Relevant current European studies on vitamin D status in vegan /vegetarian diets are missing and should also be cited.

Reviewer 3 Report

Comments and Suggestions for Authors

I think this study is meaningful as a study on the relationship between vegetarian diets and nutrition and health. However, this study is a cross-sectional study, and it is difficult to explain causality, so these limitations of the study should be mentioned in the discussion section, and the authors should be cautious in interpreting the results of the study.

1.      The conclusion in the abstract (line 23~24) is too generalized. In particular, sun exposure (the authors measured “physical activity frequency) was not analyzed in this study.  Please revise to appropriate conclusions inferred from the results of this study.

2.      Line 134~137: Please explain the physical activity survey questions in more detail. For example, whether the activity location is indoor or outdoor, or What does physical activity intensity (vigorous, moderate, walking and sitting activities) mean, etc. etc.

3.      Line 138~140:  The dietary survey was not conducted through an interview by a nutritionist or specialist but was conducted online. How can the reliability of the answers be explained? Additional explanation is needed on this. Additionally, the shortcomings of this online survey should be mentioned in the discussion section.

4.      In lines 170~173 (statistical analysis), the text "General linear models were used to test the effects of dietary group (omnivores, lacto ovo vegetarians, vegans), gender (woman, man), season, and the interaction group-gender and group-season." I don't understand the interaction for season. There are no data or results for "season" anywhere in the paper, so please add them or correct the text.

5.      lines 174~175 "The relationship between selected variables was studied by Pearson or Spearmaan correlations" - I don't know which table this refers to. Please comment below the table where this statistic was performed.

6.      In the result section, when explaining the contents of the results table, please explain the results in the order of the variables in the table. It is so confused. This applies to the contents of all tables.

7.      All table titles are inappropriate except for Table 1. Please change the titles of Tables 2 through 5 and Table 7 to reflect the content of the tables (especially Table 7 to make it clear that it is measured in blood).

8.      Lines 199~200: The text under Table 2 (lines 199-200), i.e. " The consumption of supplements apart from vitamin B12 was low, as less than 3% of 199 the total participants took calcium or vitamin D supplements'- is not included in the result table. Consider removing it or mentioning it in the discussion.

9.      Line 225~226 (Table 4): Explain why the I & Zinc content in Table 4 is presented as median rather than mean.

10.    In lines 358~359, it is questionable whether it can be concluded that the physical activity was necessarily outdoors. 

11.    It is difficult to interpret physical activity time in the same way as sunlight exposure time. This is because physical activity can be done at a sports center or at home using a treadmill. Therefore, I think explanations such as lines 358-359 “It is possible that this moderate activity was performed in leisure time and outdoors, which could explain the relationship with vitamin D status” are inappropriate. Correction is needed.

12.    Lastly, in the discussion section, please add the limitations of this study (e.g. cross-sectional study, online diet survey, etc.). Please consider that this study is a cross-sectional study and be cautious in interpreting the results. In that regard, please review the contents of the paper. Thank you for your effort.

Comments on the Quality of English Language

-

Author Response

I think this study is meaningful as a study on the relationship between vegetarian diets and nutrition and health. However, this study is a cross-sectional study, and it is difficult to explain causality, so these limitations of the study should be mentioned in the discussion section, and the authors should be cautious in interpreting the results of the study.

Yes. We agree with you on this limitation of the cross-sectional studies. This is indicated as the main limitation (lines 424-425)

  1. The conclusion in the abstract (line 23~24) is too generalized. In particular, sun exposure (the authors measured “physical activity frequency) was not analyzed in this study.  Please revise to appropriate conclusions inferred from the results of this study.

We agree with you that conclusions should be derived only from our results. We have changed the conclusion.

  1. Line 134~137: Please explain the physical activity survey questions in more detail. For example, whether the activity location is indoor or outdoor, or What does physical activity intensity (vigorous, moderate, walking and sitting activities) mean, etc. etc.

The short version of the IPAQ was used and activity intensity was asked giving the volunteers images and examples. Unfortunately, it was not asked if the activity was indoor or outdoor.

  1. Line 138~140:  The dietary survey was not conducted through an interview by a nutritionist or specialist but was conducted online. How can the reliability of the answers be explained? Additional explanation is needed on this. Additionally, the shortcomings of this online survey should be mentioned in the discussion section.

The method was 3-day dietary records. We instructed the volunteers to collect all information about food items just after they had finished their meals. This was much more useful that the interviews (that we use in case of a food frequency questionnaire). Participants were instructed on how the record all food items and they could ask doubts at any time by phone or Email. They could report on paper or on-line as we could print exactly the same questionnaires for them, but almost everybody preferred on line. Please take note that the age was 25-30 y old and they did not prefer the printed sheets. Although it is a subjective consideration, this population of vegetarians were very interested in giving correctly details of their diets.

Action taken. We added a sentence in the 2.3 paragraph: “They were required to write down this information just after finishing their meal and they could provide photos or labels of the foods they ingested and ask questions by phone or email” (lines 152-154).

  1. In lines 170~173 (statistical analysis), the text "General linear models were used to test the effects of dietary group (omnivores, lacto ovo vegetarians, vegans), gender (woman, man), season, and the interaction group-gender and group-season." I don't understand the interaction for season. There are no data or results for "season" anywhere in the paper, so please add them or correct the text.

In subheading 2.1 the settings, including year and month, were included in the original manuscript, but we agree that season was not explicitly indicated. We have added the distribution of participants according to year or recruitment, and the terms “summer” or “winter” were needed have been added. In addition, we have also included de numbers of participants, as suggested by referee 2 (lines 119-123).

  1. lines 174~175 "The relationship between selected variables was studied by Pearson or Spearman correlations" - I don't know which table this refers to. Please comment below the table where this statistic was performed.

Because the correlations only give information of the relationship between two variables, it was not included in tables but only in the text (last paragraph of the Results section). In the Statistical section, the description of the correlation study has been slightly modified (lines 194-195). As suggested by referee 2, we have added the number of participants for each correlation in the Results section (lines 299, 307).

  1. In the result section, when explaining the contents of the results table, please explain the results in the order of the variables in the table. It is so confused. This applies to the contents of all tables.

Please notice that in the revised version there is only one table with all nutrients. We have change the order or the descriptions in the Results section or the order of the nutrients listed in the table to favour reading. However, in the table the vitamins are in the order A, B, C, D.

  1. All table titles are inappropriate except for Table 1. Please change the titles of Tables 2 through 5 and Table 7 to reflect the content of the tables (especially Table 7 to make it clear that it is measured in blood).

The titles of the tables have been revised, two figures are now included in the revised version, and the table of Table 7 (Table 6 of the revised version) has been modified according to your suggestion.

  1. Lines 199~200: The text under Table 2 (lines 199-200), i.e. " The consumption of supplements apart from vitamin B12 was low, as less than 3% of 199 the total participants took calcium or vitamin D supplements'- is not included in the result table. Consider removing it or mentioning it in the discussion.

We think that this is very important because it is a very low %. We think that it should be mentioned in the results section, although it is only a level, and in the discussion/conclusion. One sentence has been added (line 372).

  1. Line 225~226 (Table 4): Explain why the I & Zinc content in Table 4 is presented as median rather than mean.

There are many results expressed as Median and 95% coefficient of variation instead of Mean with SD because the variables do not follow normal distributions. This was indicated in the statistical section (first paragraph) and in the foot of each table. Please notice, that if the variable could be normalized we applied parametric tests, but if not (eg the vitamins) we use non-parametric tests .

  1. In lines 358~359, it is questionable whether it can be concluded that the physical activity was necessarily outdoors. 

This has been deleted, also considering the comments of other reviewers

  1. It is difficult to interpret physical activity time in the same way as sunlight exposure time. This is because physical activity can be done at a sports center or at home using a treadmill. Therefore, I think explanations such as lines 358-359 “It is possible that this moderate activity was performed in leisure time and outdoors, which could explain the relationship with vitamin D status” are inappropriate. Correction is needed.

This has been deleted, also considering the comments by other reviewers

  1. Lastly, in the discussion section, please add the limitations of this study (e.g. cross-sectional study, online diet survey, etc.). Please consider that this study is a cross-sectional study and be cautious in interpreting the results. In that regard, please review the contents of the paper. Thank you for your effort.

The limitations of the study have been included.

Thank you for your revision that is has been very useful to improve our article.

Reviewer 4 Report

Comments and Suggestions for Authors

see attached file.

Author Response

This cross-sectional study investigated in young omnivores, lacto-ovo vegetarians, and vegans nutrient intake, body composition, physical activity, vitamin D status, and biomarkers of bone turnover. Body composition was similar in the three groups, but there were significantly higher PTH levels and concentrations of the bone resorption marker NTx in vegetarians than in omnivores. It is concluded that young adult vegetarians should reduce risk of bone loss by appropriate diet planning, sun exposure or vitamin D supplementation.

Generally, the manuscript is well-written and of nutritional and clinical importance. Nevertheless, there are some issues that have to be addressed.

Thank you for your opinion on our manuscript, we really appreciate it. We have addressed the issues.

  • They used a body composition monitor which is typically used in daily life, but not for scientific research. They should at least provide some validation data for the device (imprecision, accuracy; results of Bland Altman plots in comparison to gold standards of bone and muscle measurements).

Yes, we agree. Accuracy of the body weight measurement was certified by international standard procedures by ENAC (Spain) according to International Laboratories Accreditation Collaboration. Tolerance was 0.100 Kg. Precision was determined by comparing the measures of body weight, muscle mass, and body water using two monitors of the same brand and same model. The values were obtained from 6 measures, and frequency of measurement was every year. Results of precision were always <0.1%. This information has been included in the revised version (lines 135-140)

  • Describe in more detail how recruitment of study participants was done.

This has been described in more detail, considering also the suggestions by reviewer 2 (lines 115-123).

  • The hours of blood and urine sampling should be stated in the Methods section. Note that bone turnover markers show circadian variation, which may influence study results.

We agree, thank you. Participants attended our laboratory between 7:45 and 9:30 h. This has been detailed (Line 159-130).

  • The Introduction section is too long and should be shortened by one third.

We have shortened it by deleting a paragraph between lines 80 and 81 (revised version). However, we cannot reduce it because the other referees suggesting adding more information.

  • They should consider to combine Tables 3-5 to one table.

Done

  • Their conclusion of adequate sun exposure or vitamin D supplementation is not substantiated by this cross-sectional study. Delete this statement from the abstract.

Agree. Deleted

  • To become a feeling of the composition of the different diets, it would be helpful to present an additional table listing the major food components of each diet consumed by the study participants.

Thanks for your comment. We have previously collected information in terms of food items and reported associations of foods to biomarkers in vegetarians (Gallego-Narbón et al. 2019, https://doi.org/10.3390/nu11081734; Salvador et al, 2019 doi: 10.3390/nu11071659).

In the case of the present study, we decided to quantify nutrient intakes and to study the relationship of selected nutrients with bone markers.   For example, it would be very interesting to know where calcium intake of vegans comes from, and we hope we can investigate dietary habits more in depth in the near future. This is a very interesting subject.

  • Muscle and bone mass did not differ substantially between diet groups. Results are in line with the Utah paradigm of bone biology and they should refer to this relationship in the Discussion section (Schiessl et al. Bone 1998; 22:1-6).

Thanks for your suggestion. The relationship muscle-bone was mentioned in the introduction but we agree that it should be mentioned in the discussion. We have added the reference at the end of the discussion, Ref. 46 (line 417).

  • Line 29: replace ‘industrialized countries’ by ‘high-income countries’.

Thank you. Done.

  • They may wish to refer to an umbrella review indicating that low protein intake increases the risk of hip fractures (Zittermann et al. Osteoporos Int 2023).

Thank you. Yes, we mentioned the importance of proteins on bone health in the introduction. We have reinforced this idea in the discussion section (lines 416-423) and include the reference (ref. 47) that you suggested.

Round 2

Reviewer 2 Report

Comments and Suggestions for Authors

1.       (1) I think that the 3 new references are not very helpful since these studies were not designed to consider bone health. I had other studies in mind, that investigated associations between bone haellth and micronutrients in vegans and vegetarians.

2.       (3) the wording is still not unequivocally, but I expect that the authors had no clear definition like e.g. one exception a month is accepted. Therefore I accept the current version.

3.       (9) The answer " considering other lifestyle factors" line 227-228 is not sufficient. I think that the new sentence is not concrete enough. Please say, which and how these lifestile factors were considered an present these data including smoking status by group in the manuscript.

    new: the authors now conclude  in the abstract that: "Young" adult vegetarians, especially vegans, should reduce... - this reads incomprehensible since the authors do not mention the age of participants before. I suggest that you report the mean age of groups in the abstract before.

Author Response

Thank you very much for your new revision and your interest in our study. Below, you can find the answer to your points.

  1. (1) I think that the 3 new references are not very helpful since these studies were not designed to consider bone health. I had other studies in mind, that investigated associations between bone haellth and micronutrients in vegans and vegetarians.

In the original manuscript, we cited two reviews by Tucker 2014 and Falchetti et al 2022 (refs 9, 10), that include information on the relationship between micronutrients in vegan and vegetarian diets and bone health. Although many micronutrients are mentioned in these reviews, as suggested, we added the 3 new references to present publications providing specific information on selenium, zinc and iodine from vegetarian diets. Altogether, these 5 references are used to support the first two sentences of the paragraph. However, we can improve the selection of references by replacing one of them.

Action taken: We have removed ref. 12 by Hoeflich et al. that is the oldest one, and add a new one by Menzel et al. (ref 12, new version):

Menzel, J., et al. Vegan Diet and Bone Health-Results from the Cross-Sectional RBVD Study. Nutrients, 2021, 13(2), 685. https://doi.org/10.3390/nu13020685

  1. (3) the wording is still not unequivocally, but I expect that the authors had no clear definition like e.g. one exception a month is accepted. Therefore I accept the current version.

Thank you.

  1. (9) The answer " considering other lifestyle factors" line 227-228 is not sufficient. I think that the new sentence is not concrete enough. Please say, which and how these lifestile factors were considered an present these data including smoking status by group in the manuscript.

We agree that smoking is an important lifestyle factor. We have information about current smoking that was collected at time of blood sampling.

Action taken:

We have added a sentence in the Methods section: “They were asked a few questions in order to check their health status and smoking habit” (lines 160, 161 new version).

We have modified the sentence on lines 227-228 (228-230 new version), and include the estimated percentages by group: “Among other lifestyle factors, smoking was considered. It was estimated that 11%, 11% and 8% of the omnivores, lacto-ovo vegetarians, and vegans, respectively, were current smokers.

    new: the authors now conclude  in the abstract that: "Young" adult vegetarians, especially vegans, should reduce... - this reads incomprehensible since the authors do not mention the age of participants before. I suggest that you report the mean age of groups in the abstract before.

Yes, the term “young adult” is confusing. We decided to remove the word “adult” because the abstract do not give any mean±SD data.